# SGK1 Target Genes Involved in Heart and Blood Vessel Functions in PC12 Cells

**DOI:** 10.3390/cells12121641

**Published:** 2023-06-15

**Authors:** Yu-He Li, Chia-Cheng Sun, Po-Ming Chen, Hsin-Hung Chen

**Affiliations:** 1Department of Laboratory Medicine, Zuoying Branch of Kaohsiung Armed Forces General Hospital, Kaohsiung 813, Taiwan; 2Physical Examination Center, Show Chwan Memorial Hospital, Changhua 500, Taiwan; 3Research Assistant Center, Show Chwan Memorial Hospital, Changhua 500, Taiwan; 4Department of Medical Education and Research, Kaohsiung Veterans General Hospital, Kaohsiung 813, Taiwan

**Keywords:** SGK1, neuron cells, PC12 cells, RNA-seq, gene enrichment

## Abstract

Serum and glucocorticoid-regulated kinase 1 (SGK1) is expressed in neuronal cells and involved in the pathogenesis of hypertension and metabolic syndrome, regulation of neuronal function, and depression in the brain. This study aims to identify the cellular mechanisms and signaling pathways of SGK1 in neuronal cells. In this study, the SGK1 inhibitor GSK650394 is used to suppress SGK1 expression in PC12 cells using an in vitro neuroscience research platform. Comparative transcriptomic analysis was performed to investigate the effects of SGK1 inhibition in nervous cells using mRNA sequencing (RNA-seq), differentially expressed genes (DEGs), and gene enrichment analysis. In total, 12,627 genes were identified, including 675 and 2152 DEGs at 48 and 72 h after treatment with GSK650394 in PC12 cells, respectively. Gene enrichment analysis data indicated that SGK1 inhibition-induced DEGs were enriched in 94 and 173 genes associated with vascular development and functional regulation and were validated using real-time PCR, Western blotting, and GEPIA2. Therefore, this study uses RNA-seq, DEG analysis, and GEPIA2 correlation analysis to identify positive candidate genes and signaling pathways regulated by SGK1 in rat nervous cells, which will enable further exploration of the underlying molecular signaling mechanisms of SGK1 and provide new insights into neuromodulation in cardiovascular diseases.

## 1. Introduction

Serum and glucocorticoid-regulated kinase 1 (SGK1), a subfamily of serine/threonine kinases, was first cloned from rat mammary tumor cells. It is known to be transcriptionally induced by both serum and glucocorticoids [1]. SGK1 is activated by multiple factors and affects different physiological functions among various cell types and tissues [2,3]. SGK1 has been identified as a key factor in cellular stress responses and in the regulation of ion-channel transport, cell volume, glucose metabolism, hormone release, renal sodium excretion, blood pressure, cell proliferation, apoptosis, and neuronal excitability [2,4,5,6]. Abnormal SGK1 expression has been implicated in the pathogenesis of various diseases including hypertension, diabetic neuropathy, metabolic syndrome, ischemia, and neurodegenerative diseases [7,8]. In several cancers, SGK1 has been identified as a tumor-promoting gene and a key prognostic factor [9].

Previous studies have also reported the multiple roles of SGK1. In the kidney, SGK1 has been implicated in high blood pressure, salt-sensitive hypertension, renal fibrosis, type 2 diabetes, and renal inflammation. Increased SGK1 activity has been suggested as a risk factor for the development of kidney injuries [10]. The heart is a tissue with high SGK1 expression. Furthermore, SGK1 is involved in regulating blood pressure and affects the QT interval of the cardiac cycle [11,12]. During angiogenesis, SGK1, as a signaling hub, mediates protein synthesis and proliferation in endothelial cells [13]. SGK1 is also expressed in brain and nerve cells [6] and plays an important role in long-term memory formation and potentiation [14]. This kinase is a downstream mediator of the glucocorticoid-reduced neurogenesis [15]. Dysregulation of the hypothalamic-pituitary-adrenal axis (HPA) in the neuroendocrine system has been implicated in major depressive disorder (MDD), and increased glucocorticoid release from the adrenal cortex has been shown to elevate the risk of MDD [16]. SGK1 is a key mediator of HPA axis dysregulation and its overexpression has been linked to the onset of MDD [17]. SGK1 participates in the regulation of various brain functions and in the pathophysiology of various brain diseases [6].

While there have been numerous studies on SGK1 in neuronal tissues and cells, the majority of them have focused on research related to neurodegenerative diseases. Nevertheless, it is still unclear whether SGK1 has additional regulatory functions in the nervous system. A comprehensive exploration of the multiple functions of SGK1 in neurons is still lacking. Therefore, to further explore the possible role of SGK1 in regulating nerve function, this study uses SGK1 inhibitors to inhibit the expression of SGK1 in rat adrenal pheochromocytoma-derived PC12 cells, which is a commonly used in vitro research platform for neuroscience research. mRNA sequencing (RNA-seq) and differentially expressed gene (DEG) identification techniques were used to analyze transcriptome changes between GSK650394-treated and mock groups and to identify DEGs that may be regulated by SGK1. The underlying molecular signaling pathways and physiological mechanisms that may be related to SGK1 regulation were analyzed. This study contributes to the comprehensive identification of the underlying molecules and pathways that interact with SGK1 in PC12 cells.

## 2. Materials and Methods

### 2.1. Cell Line and Reagents

The rat adrenal phaeochromocytoma PC12 cell line was purchased from the Bioresource Collection and Research Center, Hsinchu, Taiwan (BCRC#60048). PC12 cells were maintained in DMEM media using 10% horse serum, 5% fetal bovine serum, 4.5 g/mL glucose, penicillin, and streptomycin. Prior to the experiments, the PC12 cell line was cultured until it reached 60–80% confluency on culture dishes coated with collagen type IV (Sigma-Aldrich, St. Louis, MO, USA; #C5533). Additionally, our research did not involve the differentiation of PC12 cells [18,19]. For SGK1 inhibition, the SGK1 inhibitor GSK650394 (Tocris Bioscience, Bristol, UK; #890842-28-1) was used. Following overnight incubation, the medium was replaced with a growth medium containing GSK650394 (100 μM) for the indicated times. The mRNA expression and protein levels of SGK1 and selected DEGs were analyzed using real-time PCR and Western blotting.

### 2.2. Real-Time PCR Assay

For RNA-seq and real-time PCR assays, after GSK650394 treatment, cellular RNA was extracted using QIAzol lysis reagent (QIAGEN, Hilden, Germany; #79306) according to the manufacturer’s instructions. The concentration and purity of the RNA were measured using a NanoDrop 8000 spectrophotometer (Thermo Scientific, Waltham, MA, USA; #ND-8000-GL). Purified RNA was then subjected to an RNA-seq analysis. For real-time PCR assays, cDNA was synthesized using 1 μg of RNA and; the Magic RT Master Mix cDNA synthesis kit (Bio-Genesis Technologies, Taipei, Taiwan; #DBU-RT-100). Real-time PCR was performed using iTaq Universal SYBR Green Supermix (Bio-Rad, Hercules, CA, USA; #1725124) with diluted cDNA and specific primers on the ABI prism StepOne Real-Time PCR system (Applied Biosystems, Waltham, MA, USA; #4376357) following the manufacturer’s instructions. The 2^−ΔΔCt^ method was used to quantify the relative levels of mRNA expression normalized to the housekeeping gene glyceraldehyde-3-phosphate dehydrogenase (GAPDH). Specific real-time PCR primers for rat *Sgk1*, selected DEGs, and *Gapdh* were designed using Primer Express software (version 3.0; Applied Biosystems, Waltham, MA, USA). The primer sequences used are listed in Table 1.

### 2.3. SDS-PAGE and Western Blotting

After GSK650394 treatment, PC12 cell lysates were harvested using the Cell Lytic M cell lysis reagent (Sigma-Aldrich, St. Louis, MO, USA; #C2978) supplemented with protease and phosphatase inhibitors (MedChemExpress, Middlesex County, NJ, USA). The concentration of cell lysates was measured using Coomassie Plus protein assay reagent (Thermo Scientific, Waltham, MA, USA; #1856210). Protein samples with Laemmli SDS sample buffers (Visual Protein, Taipei, Taiwan; #SBR06-15) were heated to 95 °C. The loaded samples were subjected to 10% SDS-PAGE and transferred to polyvinylidene fluoride membranes. The membranes were blocked with 5% skimmed milk in Tris-buffered saline with Tween 20 (TBST; 20 mM Tris-Cl, pH 8.0, 150 mM NaCl, 0.05% Tween 20) buffers at room temperature for 1 h and incubated with antibodies against SGK1 (Santa Cruz, Heidelberg, Germany; #sc-377360; 1:500), matrix metallopeptidase 9 (MMP-9) (ABclonal, Woburn, MA, USA; A11147; 1:1000), semaphorin 3C (SEMA3C) (ABclonal; A15386; 1:1000), and the internal control GAPDH (ABclonal; AC033; 1:50,000) in TBST buffers at 4 °C overnight. After washing thrice with TBST, the membrane was incubated with horseradish peroxidase-conjugated goat anti-rabbit or anti-mouse IgG (GeneTex, Hsinchu, Taiwan; #GTX213110-01 and GTX213111-01) at room temperature for 2 h. The membranes were washed thrice and visualized using an enhanced chemiluminescence system (advansta, San Jose, CA, USA; #K-12045-D50). Images were quantified using the Image Lab 6.0 software (Bio-Rad, Hercules, CA, USA) and normalized to GAPDH.

### 2.4. RNA-Seq and DEG Identification

RNA samples were qualitatively analyzed using BioAnalyzer 2100, and the RNA Integrity Number (RIN) value of all samples was >9.5. To create a sequencing library, the TruSeq Stranded mRNA library prep kit from Illumina was used. The process involved purifying mRNA from 1 μg of total RNA using oligo(dT)-coupled magnetic beads, followed by fragmentation of the mRNA into small pieces at high temperatures. Reverse transcriptase and random primers were used to synthesize first-strand cDNA, followed by generation of double-stranded cDNA and adenylation of 3′ ends of DNA fragments. Adaptors were ligated and purified using the AMPure XP system from Beckman Coulter. The quality of the resulting libraries was evaluated using the Agilent Bioanalyzer 2100 and a real-time PCR system. After assessment, the qualified libraries were sequenced on an Illumina NovaSeq 6000 platform with 150 bp paired-end reads generated by Genomics, BioSci & Tech Co. in New Taipei City, Taiwan. Then, next-generation sequencing-based RNA-seq analysis was performed (N = 1). A brief description of this method is as follows. We used fastp (version 0.20.0) trimming for low-quality sequences from adapters. Clean reads were mapped to the reference genome using HISAT2 (version 2.1.0). Feature Counts software (version 2.0.1) was used to quantify gene abundance. Gene expression levels were calculated and expressed as transcripts per million (TPM). DEGs were identified and annotated using the DESeq2 package (version 1.28.0).

DEGs were screened at an adjusted *p*-value of <0.05 and fold change (log2) >1 or <−1. A Venn diagram, volcano plot, and heatmap diagram were used to analyze the expression profiles of the selected DEGs. Venn diagram web is an intersection comparison tool and was used to compare DEGs between different groups (https://bioinfogp.cnb.csic.es/tools/venny/index.html; accessed on 18 October 2022). Upregulated and downregulated DEGs compared to the mock group were presented in volcano plots (VolcaNoseR web software, https://huygens.science.uva.nl/VolcaNoseR2/; accessed on 18 October 2022). The TPM expression profile of the top 100 DEGs in all groups took the Z-score value, and different expression changes were represented and clustered with a heatmap diagram using Pheatmap software (version 1.0.12).

### 2.5. Gene Enrichment Analysis

For gene enrichment analysis, the selected DEGs between the 72 h GSK650394-treated and mock groups were analyzed using the ShinyGO web tool (version 0.76.2, http://bioinformatics.sdstate.edu/go/; accessed on 18 October 2022). Gene functional enrichment analysis of the DEGs was performed according to gene ontology (GO) including the following three categories: biological process, cellular component, and molecular function. Pathway enrichment analysis of the DEGs was performed using the Kyoto Encyclopedia of Genes and Genomes (KEGG) database. Disease ontology enrichment analysis was performed according to the Rat Genome Database. Significantly differentially enriched GO, KEGG, and DO terms were filtered using a false discovery rate (FDR) < 0.05. Relevant radicals were used to select enriched terms related to heart and blood vessels from GO-biological process analyses and filter diseases related to cranial nervous, cardiovascular, and renal diseases.

### 2.6. GEPIA2 Correlation Analysis

GEPIA2 is an online database that facilitates the standardized analysis of RNA-seq data (http://gepia2.cancer-pku.cn/#correlation; accessed on 17 March 2023). We, therefore, employed this database to assess the link between SGK1 expression and neuromodulation in cardiovascular disease, and we further assessed the link between SGK1 expression and the expression of markers associated with left ventricle and coronary artery.

### 2.7. Statistical Analysis

Data for real-time PCR and Western blotting were obtained from three independent experiments. Data are shown as mean ± SD. Student’s *t*-test was used to determine the significance of differences between the treatment and control groups. Statistical significance was set at * *p* < 0.05, ** *p* < 0.01, and *** *p* < 0.001.

## 3. Results

### 3.1. SGK1 Inhibitor GSK650394 Inhibits SGK1 Expression in PC12 Cells

To investigate the effects of reduced SGK expression on neurophysiologic mechanisms and signaling molecules, we used an SGK1 inhibitor (GSK650394, 100 μM) in rat adrenal phaeochromocytoma PC12 cells. These cells are widely used as an in vitro research platform for physiological studies of neuronal cells [20]. Real-time PCR results showed that 62%, 69%, and 71% of *Sgk1* mRNA was significantly repressed in the 24 h, 48 h, and 72 h treatment groups, respectively (Figure 1A).

SGK1 protein levels were also detected with Western blotting (Figure 1B). SGK1 protein levels were significantly inhibited after GSK650394 treatment for 48 h (56% decrease) and 72 h (80% decrease). These results illustrate that GSK650394 could successfully reduce the RNA and protein expression of SGK1 in PC12 cells. Since treatment of PC12 cells with GSK650394 for 48 and 72 h significantly inhibited SGK1 RNA and protein expression levels, these two samples and mock samples were used for subsequent studies (these experimental groups were named GSK_48 h, GSK_72 h, and mock).

### 3.2. RNA-Seq and DEG Analysis of GSK650394-Treated PC12 Cells

To screen out potential genes that interact with SGK1, RNA-seq and DEG analyses were performed using RNA extracts of GSK_48 h, GSK_72 h, and mock to analyze transcriptome changes between the GSK650394-treated and mock groups (N = 1). The data showed that a total of 12,627 genes were sequenced and identified. Compared to the mock, the TPM of *Sgk1* was reduced by approximately 16% and 48% in the 48 h and 72 h GSK650394 treatment, respectively (Figure 2A), which is consistent with the results shown in Figure 1 that GSK650394 inhibited SGK1 expression in PC12 cells. To compare gene expression profiles with the mock, DEGs were selected with adjusted *p*-values of <0.05 and log2 fold changes of >1 or <−1. The volcano plot of DEGs showed that GSK_48 h vs. mock had 675 DEGs, including 415 upregulated and 260 downregulated DEGs, and GSK_72 h vs. mock had 2152 DEGs, including1300 upregulated and 852 downregulated DEGs (Figure 2B). Interaction Venn-diagram analyses of the two comparisons were performed. Figure 2C results revealed that of the 2239 DEGs identified, 588 DEGs were commonly observed in both GSK650394-treated cell groups (588/2239, 26.26%). However, there were higher number of DEGs in GSK_72 h vs. mock (1564/2239, 69.85%) than in GSK_48 h vs. mock (87/2239, 3.89%). RNA-seq data showed that with the increase in GSK treatment time, the fold change in the expression of related genes affected by GSK650394 treatment showed a more significant increase (fold change and *p*-value exceeding the cutoff), resulting in an increase in the number of DEGs in GSK_72 h vs. mock.

To better understand the gene expression patterns at different treatment times for GSK650394, a heatmap was used to compare the gene expression of the top 100 DEGs. The results in Figure 3 indicate two clusters. The upper cluster of DEGs was downregulated (60 DEGs), while the lower cluster was also downregulated (40 DEGs) with increasing GSK650394 treatment time. These results are similar to those in Figure 2C and suggest that the expression of more genes is affected by SGK1 inhibition by 72 h GSK650394 treatment and that these candidate DEGs related to SGK1 may play important roles in various cellular regulatory mechanisms in neuronal cells.

### 3.3. Gene Enrichment Analysis of DEGs in GSK650394-Treated PC12 Cells

We explored the possible physiological functions and regulatory mechanisms involved in these candidate DEGs. Gene enrichment analysis of 2152 DEGs from 72 h comparisons was performed using GO analysis involving three categories. Figure 4 shows the top 20 GO terms sorted by fold enrichment (FDR < 0.05) and presented as a hierarchical clustering tree. In biological processes, the DEGs were mainly enriched in blood vessel development, multicellular organismal development, cell migration, neurogenesis, and differentiation (Figure 4A). In terms of cellular components, the DEGs were predominantly enriched in neuronal cell body development and cell junctions (Figure 4B). Regarding molecular functions, the DEGs were mainly enriched in ion transmembrane transporter activity and binding of calmodulin, cytoskeletal protein, protein kinase, signaling receptors, and transcription regulatory regions (Figure 4C). In addition, KEGG pathway enrichment analysis of DEGs from 72 h comparisons was also performed and enriched in aldosterone-regulated sodium reabsorption, insulin resistance, apoptosis, and signaling pathways in Hippo, PI3K-Akt, MAPK, cancers AGE-RAGE, IL17, and TNF (Figure 4D). These results suggest that SGK1 plays a crucial role in neural development, cell growth, cell metabolism, and regulation of various signaling pathways. In particular, we found that SGK1 inhibition appears to be associated with angiogenesis regulation and development in GO-biological process analysis.

Disease ontology (DO) was developed to understand the relationship between human diseases and genes and to provide classification criteria for human diseases [21]. To further investigate the potential association of SGK1 with human diseases, we performed DO enrichment analysis of the selected DEGs between 72 h treated and mock PC12 cells. This analysis aimed to understand the relationship between human disease and DEGs related to SGK1 inhibition. The distribution of DO enrichment analysis showed that these selected DEGs were enriched in 119 DO terms (FDR < 0.05). Eight DO terms were related to brain and nervous diseases, including nervous system disease (fold enrichment: 3.01; DEGs: 14), brain neoplasms (fold enrichment: 2.84; DEGs: 12), peripheral nervous system disease (fold enrichment: 2.70; DEGs: 20), brain ischemia (fold enrichment: 2.63; DEGs: 44), transient cerebral ischemia (fold enrichment: 2.02; DEGs: 34), brain injuries (fold enrichment: 1.69; DEGs: 36), Alzheimer’s disease (fold enrichment: 1.53; DEGs: 62), and neurodevelopmental disorders (fold enrichment: 1.23; DEGs: 184). Sixteen DO terms were related to cardiovascular diseases (fold enrichment: 4.35 to 1.68 DEGs: 48 to 6), including vascular system injuries, venous thrombosis, atherosclerosis, aortic valve stenosis, myocardial ischemia and infarction, cardiac fibrosis, myocarditis, coronary artery disease, cardiomegaly, Marfan syndrome, and hypertension. Five DO terms were related to kidney diseases, including nephrosis (fold enrichment: 2.76; DEGs: 10), end-stage renal failure (fold enrichment: 2.56; DEGs: 36), insulin resistance (fold enrichment: 1.75; DEGs: 30), diabetes mellitus (fold enrichment: 1.71; DEGs: 82), and type 2 diabetes mellitus (fold enrichment: 1.55; DEGs: 71) (Figure 5). These analyses indicated that SGK1 may be involved in the functional regulation of the nervous system to the cranial nerve, cardiovascular system, and kidneys, and the absence of SGK1 may lead to the occurrence of diseases.

### 3.4. Validation of Selected DEGs Related to Heart and Blood Vessel Functions

A previous study on endothelial cells indicated that SGK1 is an important player in angiogenesis [13]. Based on the results in Figure 4A and Figure 5, the inhibition of SGK1 in PC12 cells was suggested to be associated with vascular development and cardiovascular disease. To identify DEGs involving the heart and blood in the GO-biological process DEGs list, we screened for GO terms related to the heart and blood in GO-biological process-enriched gene sets. The results indicated that 94 DEGs were enriched in 17 GO terms for the regulation of cardiac function, including heart development, cardiocyte differentiation, embryonic heart tube morphogenesis, and regulation of cardiac muscle hypertrophy in response to stress (Figure 6A). In addition, 173 DEGs were enriched in 13 blood vessel-related GO terms, including blood vessel development, regulation of blood circulation and blood pressure, regulation of blood vessel endothelial cell migration, and blood vessel endothelial cell proliferation involved in sprouting angiogenesis (Figure 6B).

To validate the accuracy of DEG expression identified from RNA-seq data, the DEGs related to heart and blood vessel regulation in Figure 6 were divided into upregulated and downregulated groups according to fold change in RNA-seq data and selected for real-time PCR. The relative expression of mRNA in the GSK_72 h group was measured, and the fold change was calculated and compared with the mock group and RNA-seq data. The results in Figure 7A indicate that all upregulated and downregulated DEGs were consistent with the trend of RNA-seq data (up: 10/10; down: 10/10). In addition, the results of blood vessel-related DEGs showed that 10 out of 11 upregulated DEGs and 10 out of 10 downregulated DEGs also showed a similar trend as the RNA-seq data (Figure 7B). We selected two DEGs with higher upregulation folds for further verification (*Mmp9* and *Sema3c*). Real-time PCR and Western blotting data indicated that both MMP-9 and SEMA3C mRNA and protein levels were significantly upregulated with increased GSK treatment time (Figure 8A,B), consistent with previous findings. The validation of real-time PCR and Western blotting results demonstrated that the results of RNA-seq and DEGs enrichment analysis were reliable and provided genetic evidence for the potential function of SGK1 in cardiovascular regulation.

Thus, we further evaluated the relationship between *Sgk1* expression and these markers in PC12 cells through the GEPIA2 database revealing similar correlations in Figure 7. Elevated *Sgk1* expression is also associated with increased cardiovascular regulation in PC12 cells, and consistent with this, the left ventricle markers *Mmp9*, *Sema3c*, and *Fgfrl1* were correlated with *Sgk1* expression. We further observed a significant correlation between *Sgk1* and markers of coronary artery including *Mmp9*, *Ccl2*, and *Anpep* (Figure 9), demonstrating that the expression level of SGK1 is positively correlated to neuromodulation in cardiovascular disease, indicating that SGK1 may have a contributing role in cardiovascular regulation of PC12, although further studies will be necessary to confirm the underlying mechanisms.

## 4. Discussion

SGK1 is widely expressed in various tissues and has been shown to mediate many important physiological functions and exert diverse roles [2,3]. Current research on the function of SGK1 in nerve cells mainly focuses on nerve growth and major depression; however, there is a lack of comprehensive functional research on whether SGK1 has other regulatory functions. To explore the underlying functions of SGK1 in nerve cells, we used an SGK1 inhibitor (GSK650394) to reduce SGK1 expression in rat PC12 cells, an in vitro neural research platform. Next, we screened SGK1 inhibition-induced DEGs by RNA-seq and DEG identification. Gene enrichment analysis was used to analyze the possible regulatory mechanisms and physiological functions of SGK1 in nerve cells. The results of this study show that SGK1 function is not only related to nerve growth and development but also that SGK1 in nerve cells may be related to cardiovascular regulation and cardiovascular diseases. The verification results of real-time PCR and Western blotting confirmed that the RNA-seq and gene enrichment data were reliable.

In this study, we successfully inhibited SGK1 expression with GSK650394 in PC12 cells and supported further RNA-seq data analysis. In the DEG analysis, more DEGs were induced at 72 h of inhibition than at 48 h of inhibition between the GSK650394-treated and mock groups. The fold change in the top 100 DEGs was time-dependent and divided into upregulation and downregulation with inhibition time (Figure 2B,C and Figure 3). Therefore, to further investigate the underlying molecular and biological processes associated with SGK1 in PC12 cells, 72 h inhibition-induced DEGs were selected for subsequent gene enrichment analysis. SGK1 inhibition induced the expression of a considerable number of genes in PC12 cells, suggesting that SGK1 interacts with these DEGs and participates in different regulatory signaling pathways.

Previous studies have shown that SGK1 is variably regulated in different cells and physiological contexts. The expression of SGK1 is upregulated after cell stress. SGK1 is activated by a signaling cascade involving phosphatidylinositol 3-kinase (PI3K), phosphoinositide-dependent kinase 1, and the mammalian target of rapamycin. SGK1 is a mediator of multiple cell membrane transports, such as ion channels and cellular glucose uptake. It stimulates transcription factors including nuclear factor-κB. During cellular stress, SGK1 regulates glucose uptake and glycolysis, angiogenesis, cell survival, cell migration, and wound healing. If cellular stress persists, SGK1 initiates fibrotic and/or calcified tissue replacement of energy-consuming cells, and cell or tissue reorganization [22,23]. It also plays a crucial role in regulating neuronal activity, proliferation, apoptosis, and memory consolidation. Increased SGK1 expression is an important risk factor for the pathogenesis of several brain diseases [6,17]. In addition, it enhances neurite formation through microtubule depolymerization [24]. The DO analysis in this study showed that SGK1 inhibition-induced DEGs in PC12 cells were associated with cranial, nervous, and renal diseases (Figure 5). The results of this study seem to be consistent with the results of the aforementioned studies, suggesting that SGK1 has a regulatory role in neuronal interactions with these DEGs and that neuronal SGK1 may be involved in the pathogenesis of these diseases.

Neuronal migration is an important process in the development of the mammalian nervous system, and abnormalities in neuronal migration can lead to neuronal migration disorders. SGK1 stimulates the migration of vascular smooth muscle cells, cancer cells, and platelets [25,26,27]. SGK1 has also been studied in relation to inflammation in helper T cells, pro-tumorigenesis in some cancer types, and as a prognostic factor [9,28]. In this study using PC12 cells, GO- and KEGG-term analysis of SGK1 inhibition-induced DEGs were also enriched for cell migration, cell junctions, cytoskeletal protein association, development of multicellular organisms, breast cancer, hepatocellular carcinoma, and IL-17 and TNF signaling pathways; however, the association of these processes with SGK1 in neuronal cells is unclear. More research is needed to confirm this, and the data from this study can be used as a reference for future research.

In the GO-biological process and DO analysis, we found that the top gene sets appeared to be related to the regulation of angiogenesis, circulatory system development, and cardiovascular disease occurrence (Figure 4A and Figure 5). SGK1 is required for embryonic vascular remodeling during angiogenesis [29]. A study using *Sgk1* KO mouse hearts found that the lack of SGK1 impairs in vitro endothelial cell tube formation and reduces blood vessel production after myocardial infarction [30]. Cardiovascular diseases often involve autonomic nervous system abnormalities, such as hypertension, cardiac diseases, arteriosclerosis, arrhythmia, myocardial lesions, and vascular lesions [19]. However, current research on whether SGK1 is involved in cardiovascular diseases is lacking. RNA-seq and DEG analysis of PC12 cells treated with corticosterone revealed that DEGs were enriched in the positive regulation of angiogenesis (GO:0045766) [31]. However, there were some limitations to our study. First, the study does not involve the differentiation of PC12 cells. Second, the RNA-seq sample size is one (Figure 2A, N = 1).

To confirm and validate our RNA-seq data, we screened the SGK1 inhibition-induced DEGs related to the heart and blood vessels in the GO-biological process (Figure 6). According to the fold changes of the RNA-seq data between the GSK_72 h and mock groups, the selected DEGs were divided into upregulated and downregulated genes and further verified using real-time PCR. As shown in Figure 7, the trend of qPCR is approximately 97.6% (40/41), consistent with the results of the RNA-seq. Most of these validated DEGs have been reported to be related to nerve cell and cardiovascular function. Among the upregulated heart-related DEGs, MMP-9 is a potential biomarker for cardiac remodeling, as demonstrated by both animal model and clinical studies [32]. Additionally, N-myc downstream-regulated gene 1 and 2 (NDRG1 and NDRG2) are substrates of SGK1 [33], and their phosphorylation is found to be impaired in SGK1−/− mice tissues [34]. Overexpression of NDRG1 reduces microvessel formation and results in a concomitant reduction in the expression of MMP-9 [35]; upregulation of NDRG2 can also significantly inhibit MMP-9 expression [36]. Thus, inhibiting SGK1 may increase the expression of MMP-9. Since MMP-9 plays a critical role in the angiogenic switch during carcinogenesis, its inhibition by NDRG1 and NDRG2expression is likely correlated with decreased angiogenesis. SEMA3C function is related to the development of spinal cord blood vessels in neuronal cells [37] and promotes angiogenesis in vascular epithelial cells [38]. Among the downregulated heart-related DEGs, G protein-coupled estrogen receptor 1 (GPER1) is associated with neuronal survival [39] and mediates protection within the cardiovascular and renal systems [40]. Among the upregulated blood vessel-related DEGs, the function of C-C motif chemokine ligand 2 (CCL2) is to enhance neuronal excitability in neuronal cells [41] and serve as a vascular permeability factor in vascular cells [42]. Among the downregulated blood vessel-related DEGs, enhanced inhibitor of DNA binding 2 (ID2) expression leads to neural stem cell failure and abnormal brain development [43]. ID2 also plays a major role in the development of hypertension in response to angiotensin II [44]. These results strongly suggest that the data obtained in this study are reliable and valuable for SGK1 research. However, there are still many unanswered questions about the associations and interactions between these DEGs and SGK1 in nerve cells and cardiovascular diseases, which, although having been studied, require further research.

## 5. Conclusions

In conclusion, this study provides reliable and valuable transcriptomic evidence for the comprehensive analysis of potential underlying candidate genes, signaling pathways, physiological mechanisms, and pathogenesis that may be regulated by SGK1 in PC12 cells. This will contribute to a better understanding of the role of SGK1 in the regulation of the nervous system and to future research on neuromodulation in cardiovascular diseases.

## Figures and Tables

**Figure 1 cells-12-01641-f001:**
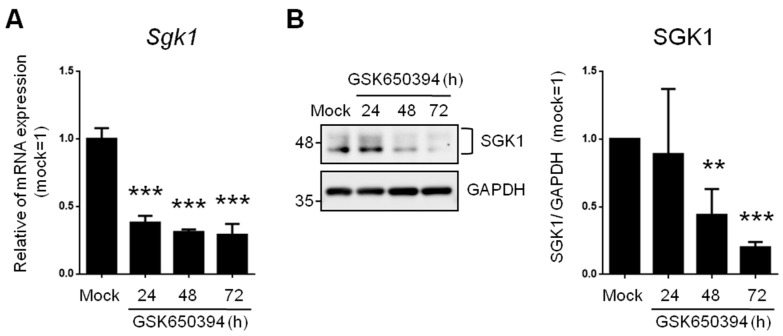
SGK1 inhibitor GSK650394 decreases gene and protein expression of SGK1 in PC12 cells. (**A**) Real-time PCR and (**B**) Western blotting of SGK1 expression in GSK650394-treated PC12 cells for indicated times. All data are shown as mean ± SD (N = 3). Statistical analysis was performed with Student *t*-test. **, *p* < 0.01; ***, *p* < 0.001 compared to the mock group.

**Figure 2 cells-12-01641-f002:**
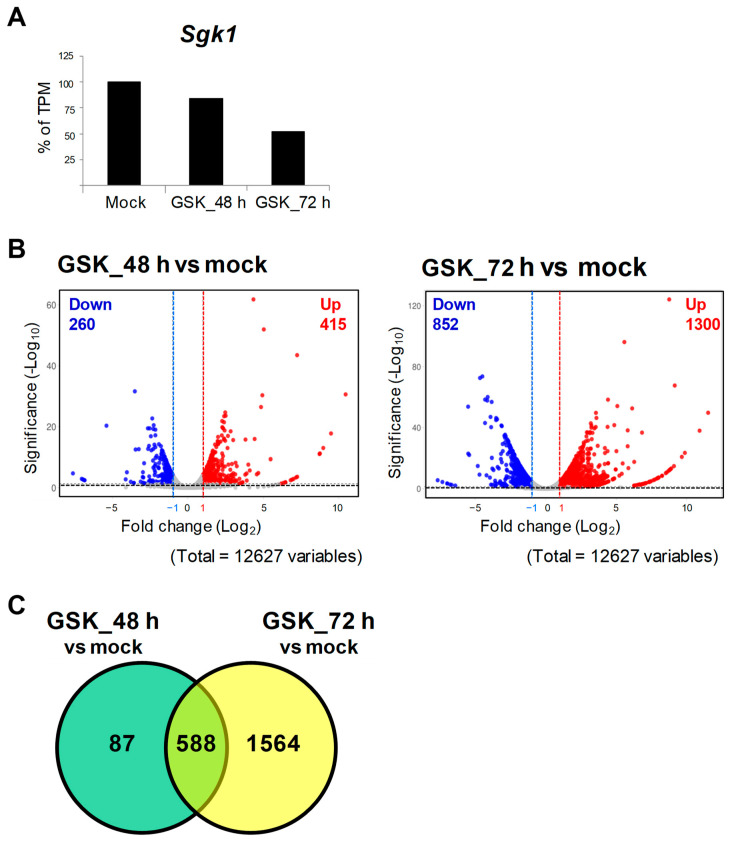
RNA-seq and differential gene expression (DEG) analysis of PC12 cells treated with GSK650394 for 48 and 72 h.(**A**) Transcript per million (TPM) of Sgk1 (N = 1). (**B**) Volcano plots of downregulated (blue) and upregulated DEGs (red). (**C**) Venn diagram of DEGs for two comparisons: GSK_48 h vs. mock and GSK_72 h vs. mock. The cutting line of DEGs is *p* adjusted < 0.05 and fold change (log2) >1 or <−1.

**Figure 3 cells-12-01641-f003:**
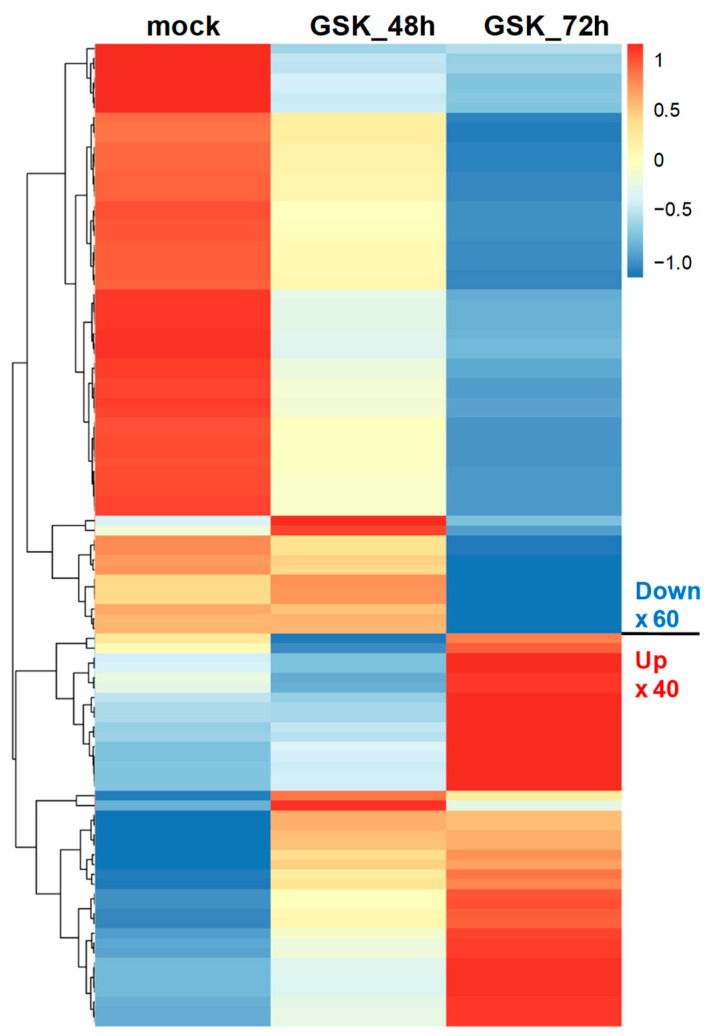
Heatmap of top 100 DEGs in PC12 cells treated using GSK650394 for 48 and 72 h. The Transcript per million (TPM) of the top 100 DEGs in all samples took the Z-score value. Different expression changes are represented by different colors, red represents high expression, and blue represents low expression. The top 100 DEGs are clustered into downregulated and upregulated clusters with increasing GSK650394 processing time.

**Figure 4 cells-12-01641-f004:**
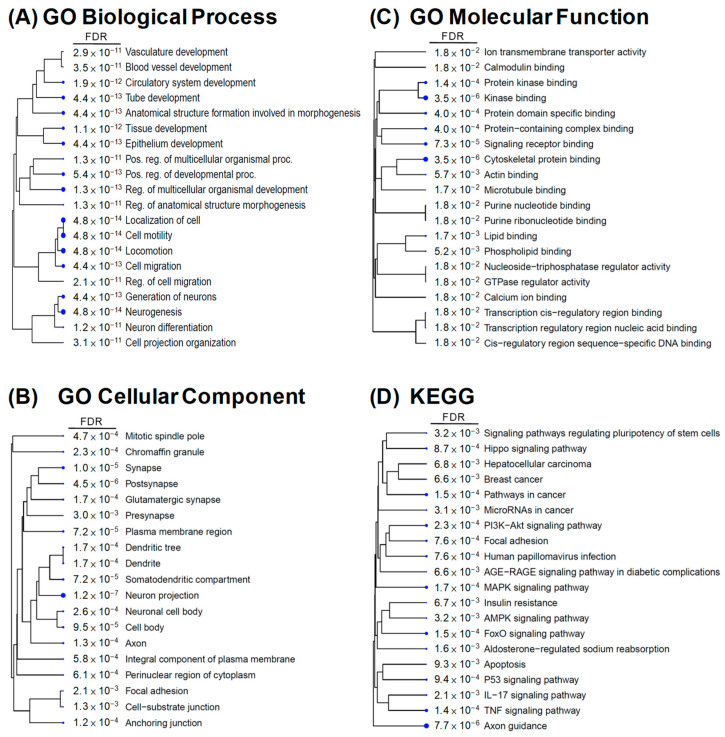
Gene enrichment analysis of DEGs from GSK650394-treated PC12 (GSK_72 h vs. mock). Gene ontology (GO) enrichment analysis for (**A**) biological process, (**B**) cellular component, (**C**) molecular function, and (**D**) KEGG pathway analysis. The top 20 GO terms or pathways are shown (FDR < 0.05). FDR is represented by numbers and blue dots. Larger dots indicate more significant FDR.

**Figure 5 cells-12-01641-f005:**
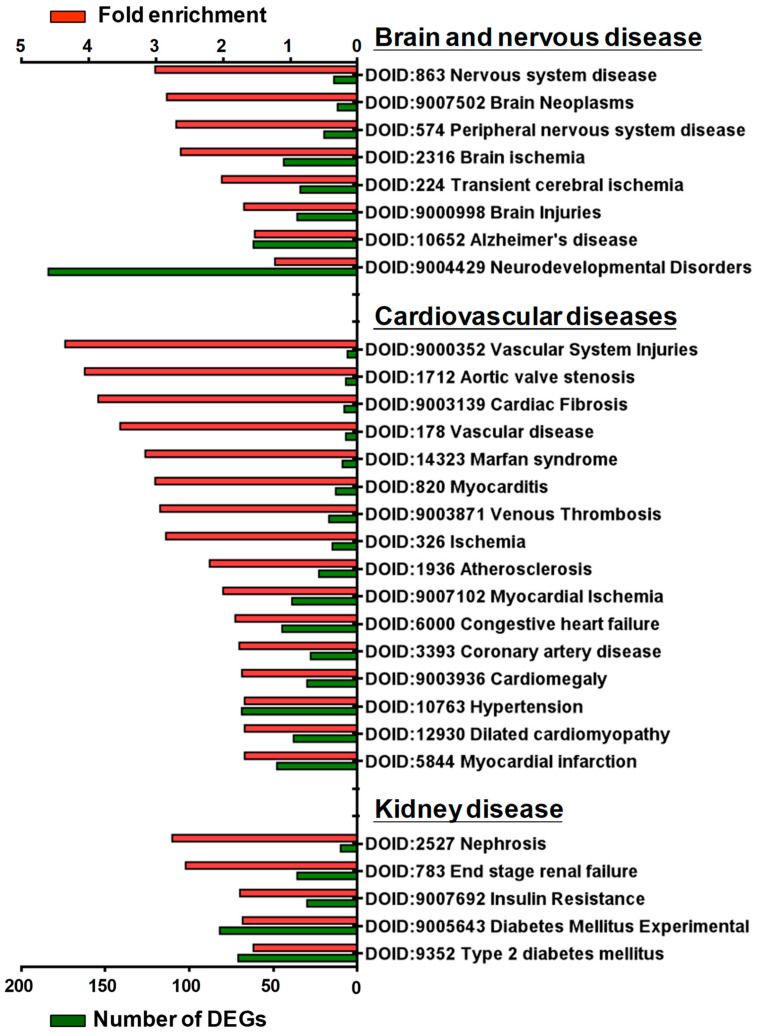
Histogram of disease ontology enrichment analysis of DEGs from GSK650394-treated PC12 (GSK_72 h vs. mock). The DEGs between GSK_72 h and mock groups were analyzed by disease ontology enrichment analysis. DO-enriched annotation terms related to brain and nervous, cardiovascular, and renal diseases were selected, and fold enrichment (red column) and numbers of DEGs (green column) involved are shown on the upper and lower parts of the x-axis, respectively.

**Figure 6 cells-12-01641-f006:**
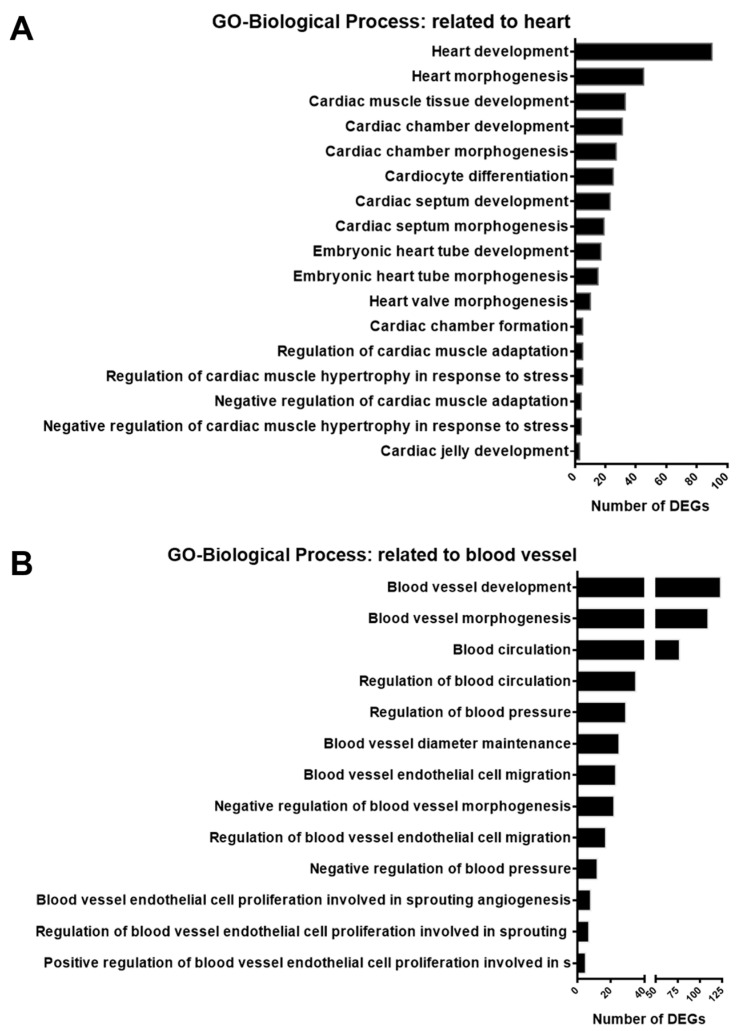
Histogram of enrichment annotations related to heart and blood vessels in gene ontology-biological process analysis of DEGs from GSK650394-treated PC12 (GSK_72 h vs. mock). The DEGs between GSK_72 h and mock groups were analyzed by gene ontology-biological process enrichment analysis. Enriched annotation terms related to the heart (**A**) and blood vessels (**B**) were selected, and the number of DEGs involved is shown on the x-axis.

**Figure 7 cells-12-01641-f007:**
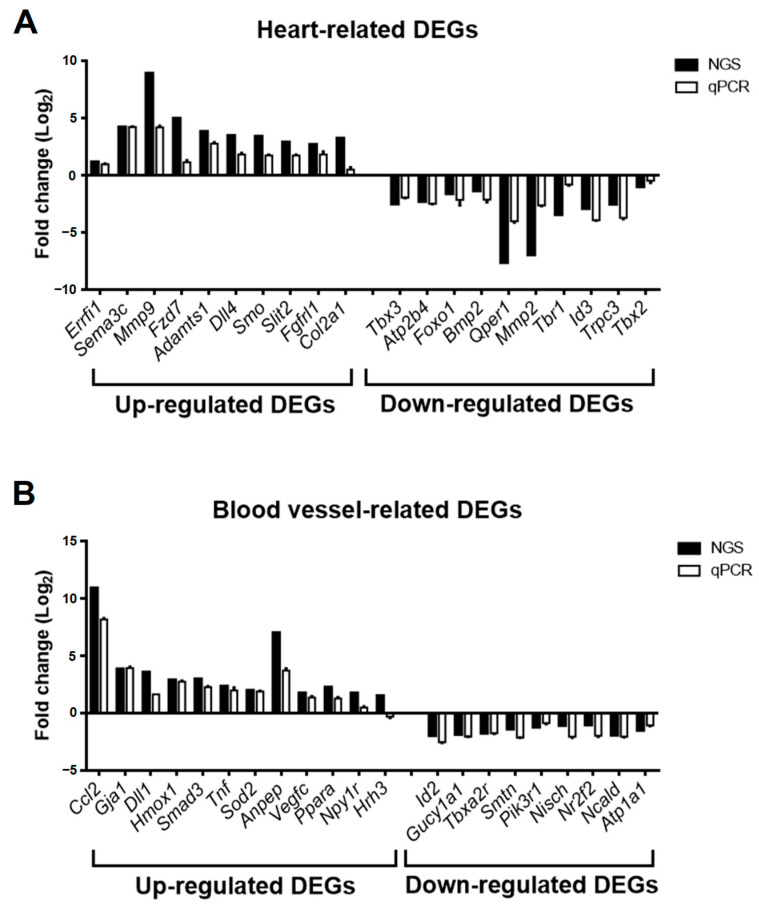
Fold change comparison of gene expression of selected DEGs from GSK650394-treated PC12 (GSK_72 h vs. mock) between RNA-seq and real-time PCR analysis. The fold change in relative gene expression of heart-related DEGs (**A**) and blood vessel-related DEGs (**B**) between GSK_72 h and the mock group are measured using real-time PCR analysis (N = 3). Fold change in the RNA-seq and real-time PCR results are represented by black and white columns, respectively.

**Figure 8 cells-12-01641-f008:**
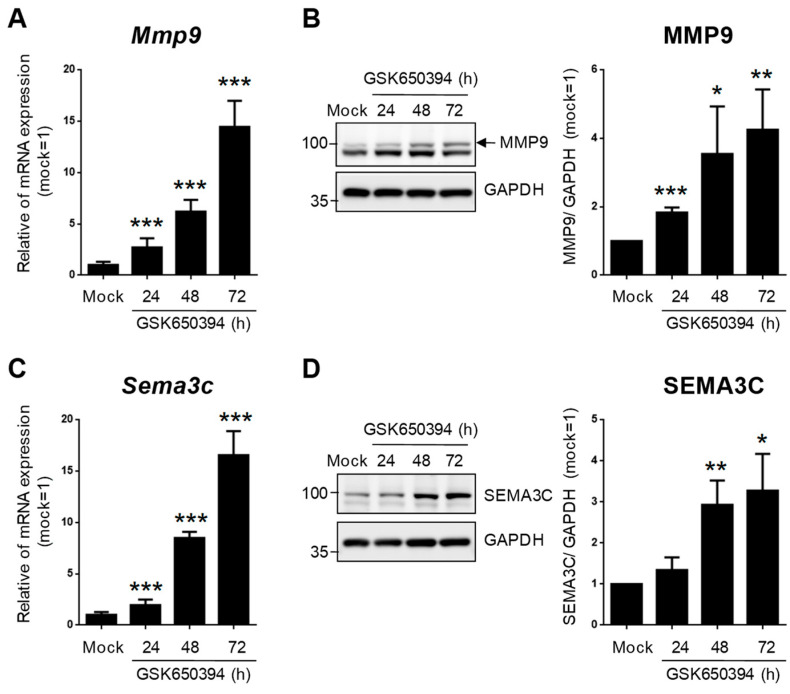
SGK1 inhibitor GSK650394 increases gene and protein expression levels of MMP-9 and SEMA3C in PC12 cells. (**A**,**C**) Real-time PCR and (**B**,**D**) Western blotting analysis of MMP-9 (**A**,**B**) and SEMA3C (**C**,**D**) expression in GSK650394-treated PC12 cells for indicated times. All data are shown as mean ± SD (N = 3). Statistical analysis was performed with Student *t*-test. *, *p* < 0.05; **, *p* < 0.01; ***, *p* < 0.001 compared to the mock group.

**Figure 9 cells-12-01641-f009:**
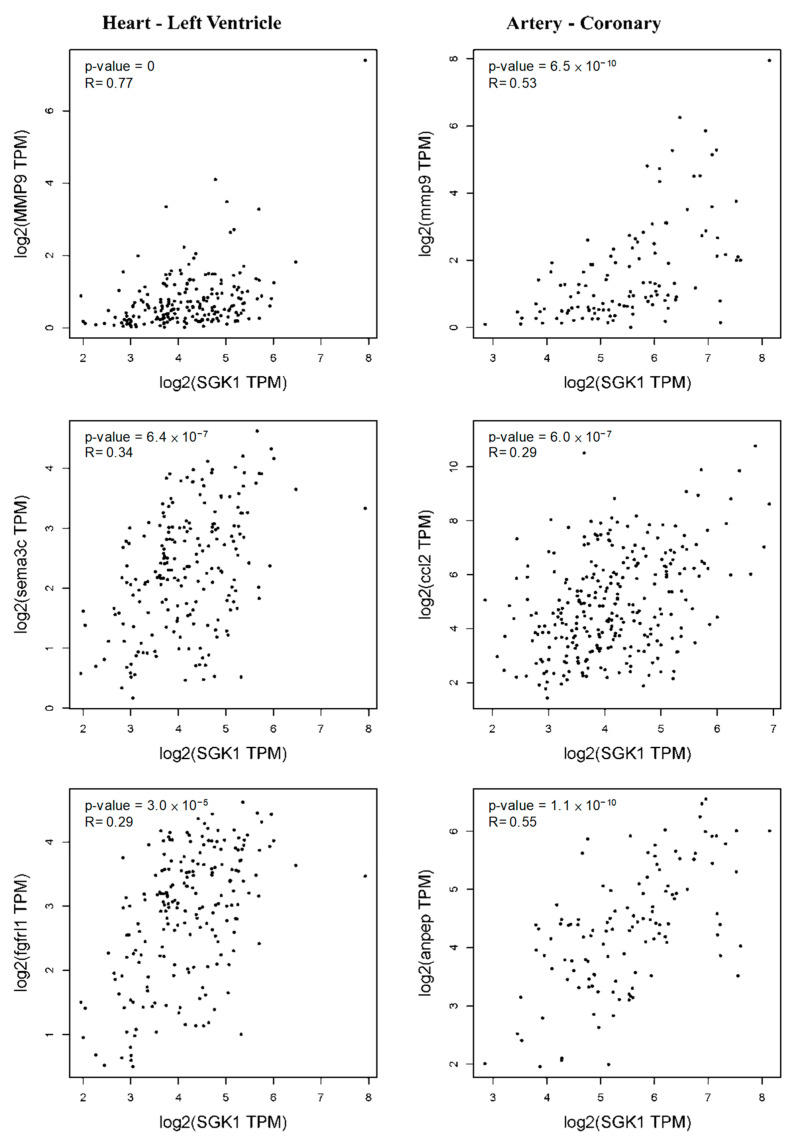
Correlation between *Sgk1* expression and markers of left ventricle and coronary artery as analyzed through GEPIA2.

**Table 1 cells-12-01641-t001:** The real-time PCR primer list.

Symbol	Description	Forward Primer Sequences (5′ > 3′)	Reverse Primer Sequences (5′ > 3′)
*Sgk1*	serum/glucocorticoid regulated kinase 1	TTCTGTGGCACGCCTGAGTA	GGTCGTACGGCTGCTTATGG
*Gapdh*	glyceraldehyde-3-phosphate dehydrogenase	GGCAAGTTCAACGGCACAGT	ATGGGTTTCCCGTTGATGAC
*Adamts1*	ADAM metallopeptidase with thrombospondin type 1 motif, 1	CAGGGCGCGGGAAAG	GGACACAAATCGCTTCTTCCTT
*Anpep*	alanyl aminopeptidase, membrane	CCAAGGGTTTCTACATTTCCAAGA	CGGCTGCCACACCTAACAG
*Atp1a1*	ATPase Na^+^/K^+^ transporting subunit alpha 1	CATCTTCCTCATTGGTATCATCGT	TGACGGTGGCCAGCAAA
*Atp2b4*	ATPase plasma membrane Ca^2+^ transporting 4	GGGCTGCAGAGTCGCATT	TGTATAAGCTGACCATTCCGGATA
*Bmp2*	bone morphogenetic protein 2	ACGACGTCCTCAGCGAGTTT	GGTCTCTGCTTCAGGCCAAA
*Ccl2*	C-C motif chemokine ligand 2	CACGCTTCTGGGCCTGTT	TGAGACAGCACGTGGATGCT
*Col2a1*	collagen type II alpha 1 chain	GACTGGCGAGTCTTGCGTCTA	CTTGCTGCTCCACCAGTTCTT
*Dll1*	delta-like canonical Notch ligand 1	CTGCACTGACCCCATTTGTCT	CCCGGTTTGTCACAATATCCA
*Dll4*	delta-like canonical Notch ligand 4	GAATAGCGGCAGTGGTCGTAA	GGTTCCCGGCCAGGTAAA
*Errfi1*	ERBB receptor feedback inhibitor 1	CGTGCCTTCCGATTTCAAA	CGCACCCACGGAAGCTT
*Fgfrl1*	fibroblast growth factor receptor-like 1	CCCACCACCGTTGACCAT	TCCAGCCACTGTGGATCGT
*Foxo1*	forkhead box O1	CCTCGAACCAGCTCAAACG	CTGCTCTGTCATGATGGGAGAA
*Fzd7*	frizzled class receptor 7	GTGCACGGATATTGCCTACAAC	CCCGCGTCCTCTTGGTT
*Gja1*	gap junction protein alpha 1	GGCCTTCCTGCTCATCCA	AGGTGTAGACCGCGCTCAA
*Gper1*	G protein-coupled estrogen receptor 1	CCTGGACGAGCAGTACTACGATATC	TGATCTGCAGGAAGAGGGACAT
*Gucy1a1*	guanylate cyclase 1 soluble subunit alpha 1	TGCTTCTCCCCGGTATCATT	TCCACGTGGCTTTCGTACAG
*Hmox1*	heme oxygenase 1	TGATGGCCTCCTTGTACCATATC	GGTTCTGCTTGTTTCGCTCTATC
*Hrh3*	histamine receptor H3	TCGCCATCTCCGACTTCCT	GTCAGCACATAGGGTACGTACAATG
*Id2*	inhibitor of DNA binding 2	CGGTGAGGTCCGTTAGGAAA	CGGGAGATGCCCAAGCT
*Id3*	inhibitor of DNA binding 3	TTAGCCTCTTGGACGACATGAA	GGACTCCCGGCACCAGTT
*Mmp2*	matrix metallopeptidase 2	ACGCTTTTCTCGAATCCATGA	ATGCTCCCATCGACCAAAGT
*Mmp9*	matrix metallopeptidase 9	TTCGAAGGCGACCTCAAGTG	TTCGGTGTAGCTTTGGATCCA
*Ncald*	neurocalcin delta	CATAGCCCTGAGTGTGACATCAA	ACATGCTGAAGGCCCACTTC
*Nisch*	Nischarin	CCCTGGAAGGCGTACACACT	AAGTTGCCTGCCAGGTTTAGG
*Npy1r*	neuropeptide Y receptor Y1	TGCTGGTCGCAGTCATGTGT	CCCAGTGGTCCATCAGTGTGT
*Nr2f2*	nuclear receptor subfamily 2 group F member 2	CAGCACCACCGCAACCA	CCCACTTTGAGGCACTTTTTG
*Pik3r1*	phosphoinositide-3-kinase regulatory subunit 1	GCAAGAGCCCTCTCTGAAATTTT	AGAGCTGGCTGCTGGGAAT
*Ppara*	peroxisome proliferator activated receptor alpha	GCTAAAGCTGGCGTACGACAA	TGTTCCGGTTCTTTTTCTGAATC
*Sema3c*	semaphorin 3C	CATTTGCGTCCAAGGATCATC	CGAAGCTCATCGAATGTTAAATACA
*Slit2*	slit guidance ligand 2	TTCCGTGGTGCAGTTGACAT	CCATCTTCAATGCAGCTGATCT
*Smad3*	SMAD family member 3	GGGCCTGCTGTCCAATGTT	AATGTGCCGCCTTGTAAGCT
*Smo*	smoothened, frizzled class receptor	ACGAGGGTGGCCTGACTTT	GGACAGCCTTCAGGGAAGTG
*Smtn*	smoothelin	AACCCCTCTTGCAGGATCATC	CGCACTCGGTCTCTCACAGA
*Sod2*	superoxide dismutase 2	GGGCTGGCTTGGCTTCA	AGCAGGCGGCAATCTGTAA
*Tbr1*	T-box brain transcription factor 1	TGCAAGGAAACCGGGTCTAT	CATCCAGTGAGCCCCAGTGT
*Tbx2*	T-box transcription factor 2	ATCGTGCGAGCCAACGA	GGAAGACATAGGTGCGAAAGGT
*Tbx3*	T-box transcription factor 3	AAGGAGACCGGAACTTCTGATG	CCTGGGCAAAGCAGTTGAA
*tbxa2r*	thromboxane A2 receptor	GCCACATGGAACCAGATCTTG	TGCAGGCGTCGAAGCA
*Tnf*	tumor necrosis factor	TGATCGGTCCCAACAAGGA	GGGCCATGGAACTGATGAGA
*Trpc3*	transient receptor potential cation channel subfamily C member 3	CCTTGGGCCTTCCATTCCT	CCCCAGCCTGCTACAAGGT
*Vegfc*	vascular endothelial growth factor C	GGCCCCAAACCAGTCACA	ACATGCACCGGCAGGAA

## Data Availability

The data presented in this study are available on request from the corresponding author.

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
