# Peer review of "SGK1 Target Genes Involved in Heart and Blood Vessel Functions in PC12 Cells"

_cells, 2023, doi:10.3390/cells12121641_

Round 1

Reviewer 1 Report

Li et al. prepared a study to identify these potential cellular mechanisms and SGK1 signaling pathways in neuronal cells.The research is mainly concerned with transcriptomic processing and various analyses of the resulting data using this high-throughput method.The whole thing represents a fairly logical sequence of events, although what I miss most are the "wet" experiments. 

The methods are described in an understandable way, but I am puzzled by the choice of research model. I would ask you to elaborate/explain: where do the inhibitor concentrations and treatment times come from? Are they chosen experimentally or based on the available literature? In trying to discover the significance of SGK1, it would not be a standard method to use gene knoukout (as has been done in other studies, even cited in the discussion).

However, the graphical rearrangement of the results is quite chaotic - no unification of the graphs (different spacing between columns, etc.). I can't see some of the statistical analyses - no error bars and significance Fig. 1C, 2A. If these significant differences are indeed not there, then we can't talk about the protein expression being lowered. It is known that the level of gene expression does not necessarily reflect protein expression. It just needs to be discussed.

Returning to figure 1 - what is the significance of including panel A in the article. I understand that the authors noticed reduced viability with the 96h treatment variant, while showing photos from under the microscope without any additional analysis does not contribute anything. Is it normal for SGK1 to come out in 2 bands during detection?

The naming of genes/proteins needs to be corrected throughout the manuscript - there are periodic rules for writing upper/lower case/cursive depending on what we are talking about (protein, gene) and the type of organism (eukaryote...). Especially note Table 1 (the authors write as if it's all the same to them).

The analysis in Fig. 2C indicated that more altered genes were noted after treatment for 72h. What is the authors' theory on this. Why these differences does time alone have that much influence? In lines 250-256 the numbers of genes should be given numerically and not in words.

Author Response

Dear editor and reviewer1:

Thank you very much for your helpful advice and numerous constructive suggestions on our manuscript titled, “SGK1 Target Genes Involved in Heart and Blood Vessel Functions in PC12 cells” (Submission ID cells-2340019) that was originally submitted on April 18, 2023. The manuscript has been revised and changes have been indicated in red, and our responses are listed below:

Thank you for your consideration. I look forward to hearing from you.

Sincerely,

Hsin-Hung Chen, PhD

https://org.vghks.gov.tw/erli/cp.aspx?n=7060F33A2B4D15F1&s=B5D2F8F17CBDA152

Department of Medical Education and Research, Kaohsiung Veterans General Hospital, Kaohsiung 813414, Taiwan

ORCID: https://orcid.org/0000-0002-5662-5945

Reviewer 1 comments

Li et al. prepared a study to identify these potential cellular mechanisms and SGK1 signaling pathways in neuronal cells. The research is mainly concerned with transcriptomic processing and various analyses of the resulting data using this high-throughput method. The whole thing represents a fairly logical sequence of events, although what I miss most are the "wet" experiments. 

Reply: Thank you for your comment. We have added some “wet” experiments. In Figure 8, we validated cardiac-associated DEG expression using real-time PCR and western blot analysis. The data indicated that GSK stimulation upregulated MMP9 and SEMA3C gene and protein expressions, consistent with RNA-seq and DEG analysis results.

The methods are described in an understandable way, but I am puzzled by the choice of research model. I would ask you to elaborate/explain: where do the inhibitor concentrations and treatment times come from? Are they chosen experimentally or based on the available literature? In trying to discover the significance of SGK1, it would not be a standard method to use gene knoukout (as has been done in other studies, even cited in the discussion).

Reply: Thank you for your comment. There are numerous ways to inhibit SGK1 expression, including KO mice, shRNA, siRNA, and SGK1 inhibitors. Among them, the SGK1 inhibitor GSK650394 is widely used in research on SGK1 and nerve cell function, so we selected GSK650394 for use in this study. In our GSK650394 (TOCRIS, 890842-28-1) preliminary experiments, we referred to some available literature and modified the methods to test the effect of different concentrations and treatment time on SGK1 inhibition in PC12 cells. The results showed that 100 μM of GSK650394 and more than 48 h of treatment time have a better inhibitory effect. Therefore, we decided to use GSK650394 as a tool to inhibit SGK1 expression and conduct follow-up studies.

However, the graphical rearrangement of the results is quite chaotic - no unification of the graphs (different spacing between columns, etc.). I can't see some of the statistical analyses - no error bars and significance Fig. 1C, 2A. If these significant differences are indeed not there, then we can't talk about the protein expression being lowered. It is known that the level of gene expression does not necessarily reflect protein expression. It just needs to be discussed.

Reply: Thank you for your comment. We have revised the figures and tables throughout the manuscript (Figures 1–9). Statistical symbols and annotations have also been added to the figures and legends (Figures 1 and 8). Since Figure 2A contains RNA-seq data (N = 1), statistical analysis could not be performed, but the results of DEG analysis through the DESeq2 package showed that the fold change of SGK1 decreased significantly (GSK_72h vs. mock: Log2 fold change = -1.12; adjusted p value = 0.00184999). GSK650394 can significantly reduce the gene and protein expressions of SGK1 as indicated in real-time PCR, western blotting, and DEG analysis. Therefore, the experimental data of this study have reference value and credibility.

Returning to figure 1 - what is the significance of including panel A in the article. I understand that the authors noticed reduced viability with the 96h treatment variant, while showing photos from under the microscope without any additional analysis does not contribute anything. Is it normal for SGK1 to come out in 2 bands during detection?

Reply: Thank you for your comment. Since Figure 1A could not clearly show the effect of GSK650394 on PC12 cell viability, we deleted it.

The SGK1 antibody (sc-377360) was obtained from Santa Cruz Company and was used for western blotting. According to the datasheet, SGK1 will have 42-, 49-, and 60-kDa bands. In this study, there were about 42- and 49-kDa bands, which is consistent with the datasheet. Part of the datasheet is attached for reference.

The naming of genes/proteins needs to be corrected throughout the manuscript - there are periodic rules for writing upper/lower case/cursive depending on what we are talking about (protein, gene) and the type of organism (eukaryote...). Especially note Table 1 (the authors write as if it's all the same to them).

Reply: Thank you for your comment. We have revised the gene and protein symbols throughout the entire paper.

The analysis in Fig. 2C indicated that more altered genes were noted after treatment for 72h. What is the authors' theory on this. Why these differences does time alone have that much influence? In lines 250-256 the numbers of genes should be given numerically and not in words.

Reply: Thank you for your comment. There were significantly more number of DEGs in the 72-h treatment. After analysis, we found that as GSK650394 treatment time increases, the impact on the expression of related genes also increases, resulting in larger fold changes and smaller p values, indicating significant differences in gene expression and increase in the number of DEGs. This same trend can also be seen in the heatmap graph (Figure 3). We have added related sentences in section 3.2. Lines 250–256 describe the DO analysis, and we have numerically presented the number of genes and fold enrichment in the relevant DO terms.

Reviewer 2 Report

The authors conducted research on the effect of SGK1 on PC12 cells using RNA sequencing analysis in their manuscript titled "SGK1 Target Genes Involved in Heart and Blood Vessel Functions in PC12 Cells". I have the following suggestions:

1. It is recommended to carefully review and edit the manuscript. For instance, in section 2.4, DESeq2 and EdgeR should be referred to as packages in Bioconductor, not as software. Additionally, it would be helpful to specify which steps of the RNA seq analysis used the DESeq2 or EdgeR package.

2. Please provide details of the RNA-sequencing protocol, such as Library Generation and sequencing platforms.

3. Traditional gene enrichment analysis focused only on differentially expressed genes (DEGs), disregarding the impact of fold change on pathway analysis. It is advisable to perform an enrichment analysis that considers both DEG p-values and fold changes.

4. For each statistical analysis, please specify the sample size (n value) in the legend section of each figure or any other appropriate sections.

5. In Figure 1C, the statistics do not match the image and increase the sample size of the WB to at least 3.

6. Please indicate the number of samples in each group for the RNAseq and subsequent analyses. The sample size of each group should be at least 3.

7. It is worth considering validating some key pathway members using Western blotting.

Author Response

Dear editor and reviewer2:

Thank you very much for your helpful advice and numerous constructive suggestions on our manuscript titled, “SGK1 Target Genes Involved in Heart and Blood Vessel Functions in PC12 cells” (Submission ID cells-2340019) that was originally submitted on April 18, 2023. The manuscript has been revised and changes are indicated in red, and our responses are listed below:

Thank you for your consideration. I look forward to hearing from you.

Sincerely,

Hsin-Hung Chen, PhD

https://org.vghks.gov.tw/erli/cp.aspx?n=7060F33A2B4D15F1&s=B5D2F8F17CBDA152

Department of Medical Education and Research, Kaohsiung Veterans General Hospital, Kaohsiung 813414, Taiwan

ORCID: https://orcid.org/0000-0002-5662-5945

Reviewer 2 comments

  1. It is recommended to carefully review and edit the manuscript. For instance, in section 2.4, DESeq2 and EdgeR should be referred to as packages in Bioconductor, not as software. Additionally, it would be helpful to specify which steps of the RNA seq analysis used the DESeq2 or EdgeR package.

Reply: Thank you for your comments. We have revised the manuscript as per your suggestion. Our RNA-seq analysis uses DESeq2 package, and the term “software” in Section 2.4 has been modified to “package.”

  1. Please provide details of the RNA-sequencing protocol, such as Library Generation and sequencing platforms.

Reply: Thank you for your comment. We have revised and provided a detailed protocol for RNA-sequencing in Section 2.4.

2.4.  RNA-seq and DEG identification

RNA samples were qualitatively analyzed using BioAnalyzer 2100, and the RNA Integrity Number (RIN) value of all samples was >9.5. To create a sequencing library, the TruSeq Stranded mRNA Library Prep Kit from Illumina was used. The process involved purifying mRNA from 1 μg of total RNA using oligo(dT)-coupled magnetic beads, followed by fragmentation of the mRNA into small pieces at high temperature. Reverse transcriptase and random primers were used to synthesize first-strand cDNA, followed by generation of double-stranded cDNA and adenylation of 3' ends of DNA fragments. Adaptors were ligated and purified using the AMPure XP system from Beckman Coulter. The quality of the resulting libraries was evaluated using the Agilent Bioanalyzer 2100 and real-time PCR system. After assessment, qualified libraries were sequenced on an Illumina NovaSeq 6000 platform with 150 bp paired-end reads generated by Genomics, BioSci& Tech Co. in New Taipei City, Taiwan. Then, next-generation sequencing-based RNA-seq analysis was performed (N = 1). A brief description of this method is as follows. We used fastp (version 0.20.0) trimming for low-quality sequences from adapters. Clean reads were mapped to the reference genome using HISAT2 (version 2.1.0). Feature Counts software (version 2.0.1) was used to quantify gene abundance. Gene expression levels were calculated and expressed as transcripts per million (TPM). DEGs were identified and annotated using DESeq2 package (version 1.28.0).

DEGs were screened at an adjusted p-value of <0.05 and fold change (log2) >1 or <–1. Venn diagram, volcano plot, and heatmap diagram were used to analyze the expression profiles of the selected DEGs. Venn diagram web is an intersection comparison tool and was used to compare DEGs between different groups (https://bioinfogp.cnb.csic.es/tools/venny/index.html). Upregulated and downregulated DEGs compared to the mock group are presented in volcano plots (VolcaNoseR web software, https://huygens.science.uva.nl/VolcaNoseR2/). The TPM expression profile of the top 100 DEGs in all groups took the Z-score value, and different expression changes were represented and clustered with a heatmap diagram using Pheatmap software (version 1.0.12).

  1. Traditional gene enrichment analysis focused only on differentially expressed genes (DEGs), disregarding the impact of fold change on pathway analysis. It is advisable to perform an enrichment analysis that considers both DEG p-values and fold changes.

Reply: Thank you for your comment. To account for the effect of fold change on pathway analysis, we screened DEGs at an adjusted p-value of <0.05 and fold change (log2) of >1 or <–1 (annotations in section 2.4). Gene enrichment analysis also set the FDR to be greater than 0.05 and focused on analyzing DEGs with large fold changes.   

  1. For each statistical analysis, please specify the sample size (n value) in the legend section of each figure or any other appropriate sections.

Reply: Thank you for your comment. We have revised and optimized the figures in this article. Statistical symbols, annotations, and sample size have also been added to the figures and legends.

  1. In Figure 1C, the statistics do not match the image and increase the sample size of the WB to at least 3.

Reply: Thank you for your comment. We have reorganized the western blotting data in Figure 1B and increased the sample size to three.

Figure 1. SGK1 inhibitor GSK650394 decreases gene and protein expression of SGK1 in PC12 cells. (A) Real-time PCR and (C) western blotting of SGK1 expression in GSK650394-treated PC12 cells for indicated times. All data are shown as mean ± SD (N = 3). Statistical analysis was performed with Student t-test. **, p < 0.01; ***, p < 0.001 compared to the mock group.

  1. Please indicate the number of samples in each group for the RNAseq and subsequent analyses. The sample size of each group should be at least 3.

Reply: Thank you for your comment. Our RNA-seq sample size is one (N = 1). The DEG analysis results through the DESeq2 package showed that the fold change of SGK1 significantly decreased (GSK_72h vs. mock: Log2 fold change = -1.12; adjusted p value = 0.00184999). GSK650394 can significantly reduce SGK1 gene and protein expressions as indicated in real-time PCR, western blotting, and DEG analysis. Therefore, the experimental data of this study have reference value and credibility.

  1. It is worth considering validating some key pathway members using Western blotting.

Reply: Thank you for your comment; we have validated the experiments using western blotting. In Figure 8, we selected two higher upregulated DEGs (Mmp9 and Sema3c) to validate cardiac-associated DEG expression using real-time PCR and western blot. The data indicated that GSK stimulation upregulated MMP9 and SEMA3C gene and protein expressions, consistent with the RNA-seq and DEG analysis results.

Reviewer 3 Report

In their article “SGK1 target genes involved in heart and blood vessel function in PC112 cells” the authors pharmacologically inhibit the kinase SGK1 and assess the effects on the transcriptome in PC12 cells. The authors carry out pathway analysis to determine the effects of SGK1 down regulation on biological function and confirm a subset of the DEGs identified by RNAseq using qPCR.

SGK1 is an important target with emerging roles in brain and disease, so this is a timely and potentially informative study.  However, there are significant concerns regarding the author’s interpretation of the data.

Gene enrichment analysis and disease ontology analysis identified nervous system, cardiovascular and kidney disease related processes, which the authors seem to interpret as a role for SGK1 in cardiac regulation. They then go on to identify and validate “heart-related DEGs”. However, this result is not supported by the analysis and their findings should be interpreted in the context of the model system used. Gene enrichment analyses like GO simply assign genes to predefined groups based on their functional characteristics; it is up to the author to interpret this data.

SGK1 downregulation results in gene expression changes that are associated with many different physiological processes. For example, many DEGs appear to be related to development (fig 4/5). It is likely that many of the same DEGs are identified across these development-pathways (vasculature, blood vessel, neuro, organismal etc). That is, it may be more accurate to interpret the DEGs as involved in developmental processes rather than in specific tissue types that are not supported by their model system. As the data are generated in PC12 cells they cannot directly support a role for SGK1 in cardiac processes. Pathway clustering approaches may provide some insight into the broader biological functions that are significantly altered following SGK1 inhibition.

Equally, the genes that the authors identify and validate as cardiac markers e.g. MMP9 and CCL2 are expressed in many other tissues and are not biomarkers of the heart per se; nor should they be – cardiac markers presumably should not be identified in PC12 cells.

The authors also attempt to link the neuro-related changes identified in their system with regulation of cardiovascular processes (lines 311-315) however, there is no data to support this – a correlation between SGK1 and MMP9 mRNA expression in PC12 cells does not support a role for neuromodulation in cardiovascular disease.

Overall, the authors revisit their interpretation of the results and edit their manuscript carefully to ensure they are not overinterpreting their data.

Other comments:

The authors should provide the method for microscopy and image capture.

Section 2.2. title states reverse transcription PCR, the methods describe real-time PCR, this should be clarified.

In section 2.3, antibody concentrations, dilutions, species and specific antibody reference #s should be provided.

The discussion should be significantly edited as it serves more as a continuation of the results e.g. lines 334-337 should be presented in the results section and lines 365-369 is a list of results but with no discussion.

Author Response

Dear editor and reviewer3:

Thank you very much for your helpful advice and numerous constructive suggestions on our manuscript titled, “SGK1 Target Genes Involved in Heart and Blood Vessel Functions in PC12 cells” (Submission ID cells-2340019) that was originally submitted on April 18, 2023. The manuscript has been revised and changes are indicated in red, and our responses are listed below:

Thank you for your consideration. I look forward to hearing from you.

Sincerely,

Hsin-Hung Chen, PhD

https://org.vghks.gov.tw/erli/cp.aspx?n=7060F33A2B4D15F1&s=B5D2F8F17CBDA152

Department of Medical Education and Research, Kaohsiung Veterans General Hospital, Kaohsiung 813414, Taiwan

ORCID: https://orcid.org/0000-0002-5662-5945

Reviewer 3 comments

  1. SGK1 downregulation results in gene expression changes that are associated with many different physiological processes. For example, many DEGs appear to be related to development (fig 4/5). It is likely that many of the same DEGs are identified across these development-pathways (vasculature, blood vessel, neuro, organismal etc). That is, it may be more accurate to interpret the DEGs as involved in developmental processes rather than in specific tissue types that are not supported by their model system. As the data are generated in PC12 cells they cannot directly support a role for SGK1 in cardiac processes. Pathway clustering approaches may provide some insight into the broader biological functions that are significantly altered following SGK1 inhibition.

Reply: Thank you for your comments. I agree to interpret the DEGs as involved in developmental processes rather than in specific tissue types that are not supported by their model system. However, previous studies have shown that molecular crosstalk between the nervous and vascular systems is necessary to maintain correct coupling of organ structure and function (PMID: 31590587, PMID: 19363295, PMID: 27920202). The results of this study show that SGK1 function is not only related to nerve growth and development but also that SGK1 in nerve cells may be related to cardio-vascular regulation and cardiovascular diseases. The verification results of real-time PCR and western blotting confirmed that the RNA-seq and gene enrichment data were reliable.

  1. Equally, the genes that the authors identify and validate as cardiac markers e.g. MMP9 and CCL2 are expressed in many other tissues and are not biomarkers of the heart per se; nor should they be – cardiac markers presumably should not be identified in PC12 cells.

Reply: Uzuelli et al. (PMID: 18047836) showed that circulating cardiac pro-MMP9 is recruited after acute pulmonary thromboembolism. In addition, Halade et al. (PMID: 23562601) showed that MMP-9 is a potential biomarker for cardiac remodeling, as demonstrated by both animal models and clinical studies on inflammation. On the other hand, we describe the mechanisms of action of CCL2 in the development and evolution of cardiovascular diseases, including heart failure, atherosclerosis and coronary atherosclerotic heart disease, hypertension, and myocardial disease.

According to your suggestion, we analyzed the results of cardiac markers, including cTn, Copeptin, PAPP-A, FABP, ST-2, and cys-C (PMID: 32016113), which were from GSK650394-treated PC12 (GSK_48h vs. mock and GSK_72h vs. mock) of RNA-seq using a heatmap as follows. The results revealed that cTn and Copeptin expressions increased in the 48-h GSK treatment in PC12 cells, but their expressions recovered at 72 h. 

  1. The authors also attempt to link the neuro-related changes identified in their system with regulation of cardiovascular processes (lines 311-315) however, there is no data to support this – a correlation between SGK1 and MMP9 mRNA expression in PC12 cells does not support a role for neuromodulation in cardiovascular disease.

Reply: Thank you for your comment; we have validated the experiments using western blotting. In Figure 8, we selected two higher upregulated DEGs (Mmp9 and Sema3c) to validate cardiac-associated DEG expression using real-time PCR and western blotting. The data indicated that GSK stimulation upregulated MMP9 and SEMA3C gene and protein expressions, consistent with the RNA-seq and DEG analysis results.

  1. Overall, the authors revisit their interpretation of the results and edit their manuscript carefully to ensure they are not overinterpreting their data.

Reply: Thank you for your comment. We have fully revised the entire content, including reorganizing the western blotting data in Figure 1B and increased sample size to three. We also validated cardiac-associated DEG expression using real-time PCR and western blotting. Our data indicate that GSK stimulation upregulates MMP9 and SEMA3C gene and protein expressions, which is consistent with our RNA-seq and DEG analysis results.

Other comments:

  1. The authors should provide the method for microscopy and image capture.

Reply: Thank you for your suggestion. Membranes were visualized using the SuperSignal™ Western Blot Substrate Bundle (A45917, Thermo Fisher Scientific, MA, USA), and images were obtained using the ChemiDocTM MP Imaging System (Bio-Rad, USA). Images were analyzed using Image Lab, version 6.0 (Bio-Rad, USA). These have been provided in the manuscript as well.

  1. Section 2.2. title states reverse transcription PCR, the methods describe real-time PCR, this should be clarified.

Reply: Thank you for your comment. We have revised Section 2.2. title and content to reflect real-time PCR.

  1. In section 2.3, antibody concentrations, dilutions, species and specific antibody reference #s should be provided.

Reply: Thank you for your comment. We have provided information on antibody concentrations, dilutions, species, and specific antibody reference numbers in section 2.3.

  1. The discussion should be significantly edited as it serves more as a continuation of the results e.g. lines 334-337 should be presented in the results section and lines 365-369 is a list of results but with no discussion.

Reply: Thank you for your suggestion. We have significantly edited the discussion section so that it no longer serves as a continuation of the results. Lines 334-337 were moved to the Results section, and lines 365-369 only list the results but do not discuss them. We have made changes based on your suggestion.

Round 2

Reviewer 1 Report

Authors adressed all concerns

Author Response

Dear Reviewer :

Thank you very much for your helpful advice and numerous constructive suggestions on our manuscript titled, “SGK1 Target Genes Involved in Heart and Blood Vessel Functions in PC12 cells” (Submission ID cells-2340019) that was originally submitted on April 18, 2023.  The manuscript has benefited from these insightful suggestions to move this manuscript closer to publication in Cells.

Sincerely,

Hsin-Hung Chen, PhD

Reviewer 2 Report

Thanks for the response and for addressing some of my concerns.

Author Response

Dear Reviewer:

Thank you very much for your helpful advice and numerous constructive suggestions on our manuscript titled, “SGK1 Target Genes Involved in Heart and Blood Vessel Functions in PC12 cells” (Submission ID cells-2340019) that was originally submitted on April 18, 2023.  The manuscript has benefited from these insightful suggestions to move this manuscript closer to publication in Cells.

Sincerely,

Hsin-Hung Chen, PhD

Reviewer 3 Report

Concerns remain regarding the authors interpretation of the data. PC12 cells are not a cardiovascular model. Pathway analysis identifies genes altered following SGK1 inhibition as being involved in cardiovascular and angiogenesis processes among others. However, PC12 cells are a cancer cell line derived from the adrenal cells and used to study aspects of neuronal function.  Although many of the DEGs in their study may be expressed in cardiac tissue, they are also expressed in other cells and tissues and the identified DOs should be interpreted accordingly. A focus on the GO findings may be more appropriate.  

The authors use the PC12 cell model to study SGK1 in a neuronal system. However, it is not clear whether the cells were differentiated prior to use as a neuron cell model and no measures of neurite length or number are provided.   

The authors also identify and validate heart-related genes like MMP9 and refer to studies e.g. Uzelli et al and Halade et al. However, these studies examined MMP9 in cardiac models or systems. MMP9 is expressed in many cells and tissues, as are the other “cardiac” genes the authors identify. With their study, the authors should discuss the association of SGK1 inhibition and MMP9 in the context of their model system.

Author Response

Dear Editor and Reviewer3:

 Thank you very much for your helpful advice and numerous constructive suggestions on our manuscript titled “SGK1 Target Genes Involved in Heart and Blood Vessel Functions in PC12 cells” (Submission ID cells-2340019), which was previously resubmitted on May 31, 2023. The manuscript has been revised and all the changes to it are indicated in red font. Our responses to all your comments are listed below.

 Thank you once again for your consideration. I look forward to hearing from you.

 Sincerely,

 Hsin-Hung Chen, PhD

https://org.vghks.gov.tw/erli/cp.aspx?n=7060F33A2B4D15F1&s=B5D2F8F17CBDA152

Department of Medical Education and Research, Kaohsiung Veterans General Hospital, Kaohsiung 813414, Taiwan

ORCID: https://orcid.org/0000-0002-5662-5945

Reviewer 3 comments

  1. Concerns remain regarding the authors interpretation of the data. PC12 cells are not a cardiovascular model. Pathway analysis identifies genes altered following SGK1 inhibition as being involved in cardiovascular and angiogenesis processes among others. However, PC12 cells are a cancer cell line derived from the adrenal cells and used to study aspects of neuronal function. Although many of the DEGs in their study may be expressed in cardiac tissue, they are also expressed in other cells and tissues and the identified DOs should be interpreted accordingly. A focus on the GO findings may be more appropriate.

Response: Thank you for your comments, which we do appreciate. We used the PC12 cell line as it is one of the most commonly used cell lines in neuroscience research. Moreover, Ieda et al. discovered that the differentiation of PC12 cells was improved when the cultures were either exposed to medium conditioned with cardiomyocytes or co-cultured with treated cardiomyocytes [1]. Additionally, when supplemented into PC12-cardiomyocyte co-culture, fibroblasts allowed long-term survival of the neurocardiac synapse [2]. Chan et al.’s findings in the ex vivo heart subjected to ischemia/reperfusion, isolated sympathetic nerve endings, and PC12 cells are compatible with what may occur in human pathophysiology [3]. In our study, several DEGs in PC12 cells were found to be related to brain function and nervous system-related, cardiovascular, and kidney diseases (Figure 5)—mainly related to the heart and blood vessels based on the GO Biological Process analysis (Figure 6). We have also further verified our main findings using immunoblotting and qPCR. Although we have not yet established a PC12-cardiomyocyte co-culture, our future research will focus on this aspect.

  1. The authors use the PC12 cell model to study SGK1 in a neuronal system. However, it is not clear whether the cells were differentiated prior to use as a neuron cell model and no measures of neurite length or number are provided.

Response: Thank you for the suggestion. Our research did not involve the differentiation of PC12 cells. However, we will consider this aspect in future studies.

  1. The authors also identify and validate heart-related genes like MMP9 and refer to studies e.g. Uzelli et al and Halade et al. However, these studies examined MMP9 in cardiac models or systems. MMP9 is expressed in many cells and tissues, as are the other “cardiac” genes the authors identify. With their study, the authors should discuss the association of SGK1 inhibition and MMP9 in the context of their model system.

Reply: Thank you for the suggestion. Accordingly, we have added the following paragraph to the Discussion:

“Among the upregulated heart-related DEGs, MMP-9 is a potential biomarker for cardiac remodeling, as demonstrated by both animal model and clinical studies [4]. Additionally, N-myc downstream-regulated gene 1 and 2 (NDRG1 and NDRG2) are substrates of SGK1 [5], and their phosphorylation is found to be impaired in SGK1−/− mice tissues [6]. Overexpression of NDRG1 reduces microvessel formation and results in a concomitant reduction in the expression of MMP-9 [7]; upregulation of NDRG2 can also significantly inhibit MMP-9 expression [8]. Thus, inhibiting SGK1 may increase the expression of MMP-9. Since MMP-9 plays a critical role in the angiogenic switch during carcinogenesis, its inhibition by NDRG1 and NDRG2 expression is likely correlated with decreased angiogenesis.” (Lines 422–432)

References:

  1. Ieda, M.; Fukuda, K.; Hisaka, Y.; Kimura, K.; Kawaguchi, H.; Fujita, J.; Shimoda, K.; Takeshita, E.; Okano, H.; Kurihara, Y.; et al. Endothelin-1 regulates cardiac sympathetic innervation in the rodent heart by controlling nerve growth factor expression. J Clin Invest 2004, 113, 876-884, doi:10.1172/JCI19480.
  2. Mias, C.; Coatrieux, C.; Denis, C.; Genet, G.; Seguelas, M.H.; Laplace, N.; Rouzaud-Laborde, C.; Calise, D.; Parini, A.; Cussac, D.; et al. Cardiac fibroblasts regulate sympathetic nerve sprouting and neurocardiac synapse stability. PloS one 2013, 8, e79068, doi:10.1371/journal.pone.0079068.
  3. Chan, N.Y.; Seyedi, N.; Takano, K.; Levi, R. An unsuspected property of natriuretic peptides: promotion of calcium-dependent catecholamine release via protein kinase G-mediated phosphodiesterase type 3 inhibition. Circulation 2012, 125, 298-307, doi:10.1161/CIRCULATIONAHA.111.059097.
  4. Halade, G.V.; Jin, Y.F.; Lindsey, M.L. Matrix metalloproteinase (MMP)-9: a proximal biomarker for cardiac remodeling and a distal biomarker for inflammation. Pharmacol Ther 2013, 139, 32-40, doi:10.1016/j.pharmthera.2013.03.009.
  5. Matschke, V.; Theiss, C.; Hollmann, M.; Schulze-Bahr, E.; Lang, F.; Seebohm, G.; Strutz-Seebohm, N. NDRG2 phosphorylation provides negative feedback for SGK1-dependent regulation of a kainate receptor in astrocytes. Front Cell Neurosci 2015, 9, 387, doi:10.3389/fncel.2015.00387.
  6. Murray, J.T.; Campbell, D.G.; Morrice, N.; Auld, G.C.; Shpiro, N.; Marquez, R.; Peggie, M.; Bain, J.; Bloomberg, G.B.; Grahammer, F.; et al. Exploitation of KESTREL to identify NDRG family members as physiological substrates for SGK1 and GSK3. Biochem J 2004, 384, 477-488, doi:10.1042/BJ20041057.
  7. Maruyama, Y.; Ono, M.; Kawahara, A.; Yokoyama, T.; Basaki, Y.; Kage, M.; Aoyagi, S.; Kinoshita, H.; Kuwano, M. Tumor growth suppression in pancreatic cancer by a putative metastasis suppressor gene Cap43/NDRG1/Drg-1 through modulation of angiogenesis. Cancer Res 2006, 66, 6233-6242, doi:10.1158/0008-5472.CAN-06-0183.
  8. Kim, A.; Kim, M.J.; Yang, Y.; Kim, J.W.; Yeom, Y.I.; Lim, J.S. Suppression of NF-kappaB activity by NDRG2 expression attenuates the invasive potential of highly malignant tumor cells. Carcinogenesis 2009, 30, 927-936, doi:10.1093/carcin/bgp072.